# Synthesis, Crystal Structure, and Characterization of Energetic Salts Based on 3,5-Diamino-4*H*-Pyrazol-4-One Oxime

**DOI:** 10.3390/molecules28010457

**Published:** 2023-01-03

**Authors:** Wen-Shuai Dong, Lu Zhang, Wen-Li Cao, Zu-Jia Lu, Qamar-un-Nisa Tariq, Chao Zhang, Xiao-Wei Wu, Zong-You Li, Jian-Guo Zhang

**Affiliations:** 1State Key Laboratory of Explosion Science and Technology, Beijing Institute of Technology, Beijing 100081, China; 2China Ordnance Society, Beijing 100089, China

**Keywords:** 3,5-diamino-4*H*-pyrazol-4-one oxime, energetic salts, crystal structures, thermal analysis

## Abstract

In order to broaden the study of energetic cations, a cation 3,5-diamino-4*H*-pyrazol-4-one oxime (DAPO) with good thermal stability was proposed, and its three salts were synthesized by a simple and efficient method. The structures of the three salts were verified by infrared spectroscopy, mass spectrometry, elemental analysis, and single crystal X-ray diffraction. The thermal stabilities of the three salts were verified by differential scanning calorimetry and thermos-gravimetric analysis. DAPO-based energetic salts are analysed using a variety of theoretical techniques, such as 2D fingerprint, Hirshfeld surface, and non-covalent interaction. Among them, the energy properties of perchlorate (DAPOP) and picrate (DAPOT) were determined by EXPLO5 program combined with the measured density and enthalpy of formation. These compounds have high density, acceptable detonation performance, good thermal stability, and satisfactory sensitivity. The intermolecular interactions of the four compounds were studied by Hirshfeld surface and non-covalent interactions, indicating that hydrogen bonds and π–π stacking interactions are the reasons for the extracellular properties of perchlorate (DAPOP) and picrate (DAPOT), indicating that DAPO is an optional nitrogen-rich cation for the design and synthesis of novel energetic materials with excellent properties.

## 1. Introduction

2,4,6-trinitrotoluene (TNT), 1,3,5-trinitro1,3,5-triazine (RDX), and Hexanitrohexaazaisowurtzitane (CL-20) are well-known traditional energetic materials, and their energy is mainly derived from the oxidative decomposition of carbon skeletons with significant ring tension and a large number of nitro groups, which is difficult to avoid their high sensitivity to mechanical and thermal stimuli while having high density [1,2,3]. The development of new high energy-density materials (HEDMs) with practicality, safety, and environmental friendliness has always been the focus of research in the field of energetic materials, and these are the key points in the development of azole containing energetic ionic salts [4,5,6,7]. Through electrostatic attraction and hydrogen bonding, azole energetic ionic salts tightly combine anions and cations with more N-N and C-N bonds, so that they have high enthalpy of formation and thermal stability, better energy characteristics, and oxygen balance [8,9,10,11]. The structure of organic ion salts does not contain toxic heavy metal ions, and their decomposition produces a large number of non-toxic and non-polluting gases such as nitrogen, which is environmentally friendly and can be widely used in explosives, gas generators, pyrotechnics, and other fields [12,13].

As an important branch of energetic materials, polyazole-based energetic ionic salts have advantages and are better in terms of properties than similar non-ionic energetic compounds, gaining much attention in the current field of energetic materials [14,15,16]. Energetic salts have lower vapor pressure, better thermal stability, higher density, and enthalpy of formation [17,18]. In addition, the properties of energetic salts can be comprehensively adjusted and optimized through the modification and combination of anions and cations, making them suitable for different applications under various conditions, and at the same time, the types of energetic ionic salts can be greatly increased simultaneously [19,20]. The azole energetic salts not only retain the high tension of the azole ring, but they also contain the above-mentioned advantages of energetic ionic salts with a high enthalpy of formation [21]. The superior properties of such functional materials expand the application of energetic materials.

In this work, 3,5-diamino-4*H*-pyrazol-4-one oxime (DAPO) was used as a nitrogen-rich cation to synthesize its hydrochloride (DAPOC), perchlorate (DAPOP), and picrate (DAPOT). The three energetic salts were comprehensively analyzed and characterized by infrared spectroscopy, mass spectrometry, elemental analysis, single crystal X-ray diffraction, differential scanning calorimetry (DSC), and thermogravimetric analysis (TG). The crystal structures were theoretically analyzed by Hirshfeld surface, two-dimensional fingerprint, and non-covalent interaction to explore the relationship between the structural characteristics of compounds and the thermal stability or sensitivity properties of molecules. The detonation performances of perchlorate (DAPOP) and picante (DAPOT) were calculated using EXPLO5 software, and their sensitivities were studied using the BAM method.

## 2. Results

Using malononitrile as raw materials, 3,5-diamino-4*H*-pyrazol-4-one oxime (DAPO) was synthesized by a three-step reaction [22,23], and the synthetic route of the three salts is in Figure 1:

### 2.1. X-ray Structure Analysis

The crystal structures of DAPO and its three salts were confirmed by X-ray single crystal diffraction at 293 K and 163.15 K. The detailed crystallographic data for the four crystal structures were listed in Table 1 and Appendix A, where some bond length, bond angle, and dihedral angle data are placed in the Support Material. Some common characteristics can be observed in these four energetic compounds. For example, in the crystal structures of DAPO and its three energetic salts, the bond length of the C–N bond on the pyrazole ring is in the range of 1.310–1.329 Å, which is between the C=N (1.20 Å) and the C–N (1.47 Å), and the DAPO molecules in all four structures are planar structures, indicating the presence of multi-bond features and the π–π conjugation effects in these crystal structures.

DAPO crystal belongs to P2_1_/c space group of monoclinic system, with four molecules per unit cell (Z = 4) (Figure 1a). The C–N bond of the pyrazole ring (C1–N2, 1.310(4) Å; C3–N1, 1.329(4) Å) are longer than C-N bond from the C-amino groups (C3–N5, 1.330 (4) Å; C1–N3, 1.361 (4)). DAPO is almost a flat structure, proved by the dihedral angle of N2–N1–C3–N5 (179.2°), N2–C1–C2–N4 (177.9°), O1–N4–C2–C3 (177.9°), and N1–N2–C1–N3 (178.7°), respectively. As shown in Figure 1b,c, on the same layer, adjacent molecules are linked in a head-tail pattern by two sets of strong hydrogen bonds (N3−H3B…O2 2.580 Å, N1−H1…O3 1.990 Å) with H_2_O as bridge and in the form of side by side by hydrogen bonds composed by the hydrogen atom on the pyrazole and the amino group on the adjacent pyrazole (N5−H5B…N2 2.150 Å), formed one-dimensional structure.

DAPOC crystallizes in the monoclinic space group Cc with two molecules per unit cell (Z = 2) with a crystal density of 1.548 g·cm^−3^. Figure 2b shows the 2D layered structure and the mesh hydrogen-bond structure of the crystal, which can be explained by the strong hydrogen bond (N–H…Cl) and (N–H…O) as the center of the network to make the structure tighter, which is favorable to reduce sensitivity. Each H_2_O molecular interacts with adjacent chloride ions and two DAPO through strong hydrogen bonds (O2−H2C…Cl1 2.330 Å, N1−H1B…O2 2.200 Å, and N2−H2B…O2 2.011 Å) to construct a crossing crystal stack as shown in Figure 2b,c.

DAPOP crystallizes in the triclinic space group *P*-1 with four molecules per unit cell (Z = 4) with a crystal density of 1.703 g·cm^−3^ (Figure 3a). As shown in Figure 3b, three adjacent DAPO cations are linked in the head-to-tail mode by hydrogen bonds on the same layer. (N3−H3A…O8 2.370 Å, N3−H3B…O10 2.209 Å, and N8−H8A…O7 2.119 Å) using the perchlorate as the bridge. Figure 3c shows that the crystal structure of DAPOP is stacked in the form of layers, which is due to the fact that perchlorate anions are filled in the space formed by cations, and DAPO cations are connected to perchlorate by strong intramolecular hydrogen bond. At the same time, layers are connected by hydrogen bonds (O7−H11B...O11 2.242Å) to form a stable 3D structure. The distance between layers is 3.0849Å, which is smaller than the typical aromatic face-to-face π–π interaction (3.40Å), indicating that there is strong π–π stacking among the crystal layers. Furthermore, the crystal density is generally related to the detonation performance of energetic materials. DAPOP has the highest crystal density among the three ionic salts, with a crystal density of 1.703 g·cm^−3^ at 293 (2) K, which may be related to stronger π–π interaction in crystal structure.

DAPOT crystallizes in the orthorhombic triclinic space group *Pbcm* with four molecules per unit cell (Z = 4) with a density of 1.701 g·cm^−3^ (Figure 4a). As shown in Figure 4b, three adjacent cations are bonded by hydrogen bonds (N5−H5A…O1 2.035 Å, N5−H5A…O2 2.191 Å, N3−H3A…O4 2.225 Å, O9−H9A…O3 2.079 Å, O11−H11…O8 1.813 Å, N5−H5A…O1 2.035 Å, and N2−H2…O8 1.891 Å) on the same layer. Figure 4c shows the layered crystal stack of DAPOT·H_2_O, which uses H_2_O as a bridge to stabilize the 3D structure through strong intramolecular hydrogen bond (O−H...O). The layers are stacked into a layered structure with a distance of 3.1695 Å, which can be regarded as a typical structure according to the face-to-face π–π interaction (<3.40 Å) [24].

### 2.2. Intermolecular Interactions

To understand the π–π interaction and hydrogen bond distribution of these compounds, the Hirshfeld surface, 2D fingerprint, and the percentage distribution of each atom were analyzed by CrystalExplorer, as shown in Figure 5a–d. Hirshfeld surface analysis has been proven to be a useful method for calculating and analyzing intermolecular interactions in crystal structures. The red and blue areas on the Hirshfeld surface denote high and low close-contact portions, respectively. Influenced by the coplanar conjugated configuration of the cation, many red patches distributing at the edge of the flat plate-like Hirshfeld surface of the cation were observed in salts DAPO·2H_2_O, DAPOC·H_2_O, DAPOP·H_2_O, and DAPOT·3H_2_O, indicating that the existence of strong hydrogen bonds in these crystal structures. The red and blue regions on the surface of Hirshfeld surface indicate higher and lower close contact between molecules, respectively. Due to the planar structure of DAPO, massive red patches can be observed at the edge of the surface of DAPO·2H_2_O and its energetic salts DAPOC·H_2_O, DAPOP·H_2_O, and DAPOT·3H_2_O, indicating that in the presence of weak interactions or strong hydrogen bonds in these crystal structures. As can be analyzed from the 2D fingerprint, two pairs of significant spikes at the lower left represent the presence of hydrogen bonds O...H and N...H, as shown in Figure 5e–h. From the pie chart, the contribution percentage of the atomic-to-atomic contact of these compounds to the Hirshfeld surface can be determined. As shown in Figure 5i–l, O…H and N…H accounted for 51% of the total weak interactions for DAPO·2H_2_O, 62% for DAPOC·H_2_O, 40% for DAPOP·H_2_O, and 50% for DAPOT·3H_2_O, respectively. The H…Cl contact accounted for 11% of the total weak interactions for DAPOP·H_2_O. All these strong interactions showed that the intermolecular stability is mainly maintained by ionic bonds, π–π interaction forces, and hydrogen bonds.

For polynitrogen heterocyclic energetic compounds, the π−π stacking is represented by C−O, C−N, N−N, and O−N interactions. This can be seen by 2D fingerprint analysis, DAPOT·3H_2_O has the largest interaction percentage (24%) compared with others (13% in DAPO·2H_2_O, 9% in DAPOC·H_2_O, and 10% in DAPOP·H_2_O), which shows that there is a high interlayer contact between the layers of the structure, reflecting the strong interaction between adjacent layers. Non-covalent interaction (NCI) diagrams of graded surfaces can be used as complementary calculations for weak interactions. By analyzing the reduced density gradient (RDG) isosurface plot of the colors-filled in the NCI plot, the colors of which can be easily and efficiently analyzed to identify different types of weak interaction regions [25,26]. As shown in Figure 6, hydrogen bonds can be recognized as small blue ellipses. As the lots of accumulation of electron density increases, the dark blue region implies a strong hydrogen bonding. From this analysis, an obvious blue region between H atom on cation and O and N atom from anion can be observed, indicating a strong hydrogen bond interaction. At the same time, due to the tightly parallel structure between layers (Figure 6a–d), π–π interactions can be easily observed between two adjacent molecular layers, as shown in Figure 6e–h, which can enhance molecular packing stability and reduce the sensitivity of energetic compounds.

### 2.3. Energetic Performance and Safety

For energetic materials, detonation performance and safety performance are the most important performance parameters, and were listed in Table 2. The DAPOP and DAPOT samples were dried in a vacuum environment at 90 °C for 48 h to guarantee that there is no crystal water before the density test. The densities of DAPOP and DAPOT are 1.831 g·cm^−3^ and 1.819 g·cm^−3^, respectively, higher than that of RDX (1.80 g·cm^−3^), which were measured at room temperature using a densimeter. The standard molar enthalpies of formation (Δ_f_*H*^θ^_m_) of DAPOP and DAPOT were calculated using the Gaussian 09 program [27]. The detonation characteristics of DAPOP and DAPOT were calculated by the EXPLO5 (v6.04) program based on the standard molar enthalpy of formation (Δ_f_*H*^θ^_m_) and experimental density at room temperature. The detonation velocities of DAPOP and DAPOT are 8249 m·s^−1^ and 7865 m·s^−1^, respectively. The detonation pressures are 25.9 GPa and 30.0 GPa, respectively, showing acceptable detonation parameters. In addition, the energetic salts based on DAPO were compared with the energetic salts of 4-oxyl-3,5-dinitropyrazolate (DNPO). DAPO-based energetic salts have higher density and enthalpy of formation, where DNPO-based energetic salts have good detonation performance due to their nitro and hydroxyl functional groups. DAPOT has a high decomposition temperature, low mechanical sensitivity, and acceptable energy performance, and is an insensitive energetic material with excellent comprehensive performance.

The thermal stability and sensitivity of the compounds were tested to evaluate the safety of the compound. At a heating rate of 10 °C·min^−1^, their thermal stability was determined with DSC and TG. The four compounds show good thermal stability, and the decomposition temperature is between 182 °C and 288 °C, as shown in Figure 7. Except for the energetic salt DAPOP·H_2_O (177 °C), the thermal decomposition temperatures of others all exceed 220 °C. Among them, decomposition peak temperature of DAPO·2H_2_O is 288 °C, showing good thermal properties, higher than RDX and HMX. For DAPOC·H_2_O, the DSC curve shows an endothermic and exothermic process. The endothermic process at 130 °C corresponds to 11% weight loss at 110−135 °C in the TG curve. During the exothermic process, the decomposition peak is 226 °C, and the weight loss is 47% in the TG curve of 206−240 °C. Since the anion of DAPOC·H_2_O is not energetic ion, its exothermic peak is gentler and releases less energy than other energetic salts based on DAPO. As for the other two energetic salts (DAPOP·H_2_O and DAPOT·3H_2_O), an exothermic process occurred unexpectedly in the DSC curve at 182.0 °C for DAPOP·H_2_O, corresponding to a weight loss process in the TGA curve at 174−187 °C, with a weight loss of 43%, showing a serious mass loss process. DAPOT·3H_2_O has one endothermic and two exothermic processes. The endothermic process at 96 °C corresponds to 12% weight loss in the TG curve. The first exothermic decomposition peak is 225 °C, weight loss is 44%, and the second decomposition peak is 293 °C, weight loss is 20%. The thermal stability of DAPOT·3H_2_O is higher than that of DAPOP·H_2_O is because DAPOT·3H_2_O has more hydrogen bonds and stronger π−π interactions. The impact and friction sensitivity were tested by standard BAM drop hammer and BAM friction tester respectively. DAPOP and DAPOT have low mechanical sensitivity (IS: 24−40 J; FS: 224−360 N) due to their layered stacking structures.

## 3. Materials and Methods

Although we do not encounter any danger when dealing with these materials, especially DAPOP and DAPOT. We strongly encourage the use of small-scale, safe synthetic methods (face shields and leather gloves). All the reagents used in the article are reagent grade and can be used without further treatment. Elemental analyses were performed on a Flash EA 1112 fully automatic trace element analyzer (Waltham, MA, USA). The FT-IR spectra were recorder as KBr pellets on a Bruker Equinox 55 (Bruker, Germany). Mass spectra were recorded on an Agilent 500-MS (Palo Alto, CA, USA). The single-crystal X-ray diffraction analysis was carried out by on Bruker CCD area-detector diffractometer (Bruker, Bremen, Germany).

### 3.1. 3,5-diamino-4H-pyrazol-4-one Oxime (DAPO)

Under the condition of ice water bath, malononitrile (6.5 g) was added to the mixed solution of acetic acid (9.6 mL) and potassium acetate (0.294 g) in 100 mL water, which was slowly added to the above solution in batches, and then heated to room temperature for 12 h. The orange solid was obtained by spin-drying the solvent. The orange solid was mixed with hydrazine hydrochloride (5.0 g) and dissolved in 60 mL water. The solution became black after standing for 48 h. The pH was adjusted to 7–8 with sodium bicarbonate, and the deep red solid was obtained by filtration. The yield is 70%. DSC (10 °C·min^−1^): 286 °C (onset). IR (ν, cm^−1^): 3419, 3178, 1675, 1623, 1576, 1506, 1419, 1229, 1106, 997, 833, 776, 731, 653.

### 3.2. 3,5-diamino-4H-pyrazol-4-one Oxime Hydrochloride Salt (DAPOC)

DAPO (127 mg, 1 mmol) was suspended in 15 mL distilled water, and then 3 mL 20% hydrochloric acid was slowly added under stirring conditions. The solution was further stirred for 1 h, and crystallized at room temperature by solvent evaporation. After 2 days, black-red bulk crystals were collected. The yield is 84%. DSC (10 °C·min^−1^): 206 °C (onset). MS (ESI^+^): 128.06 [C3H6N5O^+^]. IR (ν, cm^−1^): 3118, 1701, 1664, 1628, 1580, 1457, 1058, 991, 903, 872, 860, 842, 767, 733, 715, 704, 686, 663, 643, 636, 629.

### 3.3. 3,5-diamino-4H-pyrazol-4-one Oxime Perchlorate Salt (DAPOP)

DAPO (127 mg, 1 mmol) was dissolved in 10% HClO_4_ solution (15 mL) and stirred at 75 °C for 45 min. The transparent solution was slowly volatilized at room temperature, and a red needle crystal was obtained after 3 days. The yield is 80%. DSC (10 °C·min^−1^): 286 °C (onset). MS (ESI^+^): 128.06 [C_3_H_6_N_5_O^+^], MS (ESI^−^): 98.95 [ClO_4_^−^]. IR (ν, cm^−1^): 3282, 1653, 1624, 1419, 1232, 1066, 935, 907, 893, 852, 838, 776, 734, 720, 705, 686, 677, 669, 656, 638, 624.

### 3.4. 3,5-diamino-4H-pyrazol-4-one Oxime Picrate (DAPOT)

Picric acid (229 mg, 1 mmol) was slowly added in batches in the suspension of DAPO (127 mg, 1 mmol) and distilled water (15 mL). The mixture is then heated to 80 °C and stirred until transparent. The filtrate volatilizes slowly at room temperature. After 6 days, red bulk crystals were collected. The yield is 78%. DSC (10 °C·min^−1^): 205 °C. MS (ESI^+^): 128.06 [C_3_H_6_N_5_O^+^], MS (ESI^−^): 227.99 [C_6_H_2_N_3_O_7_^−^]. IR (ν, cm^−1^): 3168, 1701, 1648, 1594, 1580, 1473, 1430, 1364, 1332, 1315, 1266, 1157, 1132, 1065, 980, 930, 907, 787, 761, 742, 709, 668, 635.

## 4. Conclusions

Generally, the synthesis of three salts (hydrochloride (DAPOC), perchlorate (DAPOP), and picrate (DAPOT)) were performed successfully and characterized based on 3,5-diamino-4*H*-pyrazol-4-one oxime (DAPO). Their crystal structures were texted by single crystal X-ray diffraction, and then the weak interactions and hydrogen bonds between their structures were studied by simulation calculations. Planar conjugated cations, rich hydrogen bonds, and extensive π–π interactions result in good physicochemical properties and detonation properties, especially for high-energy salts DAPOP and DAPOT. The decomposition temperature of the three energetic salts is between 182 and 226 °C, and the thermal stability of DAPOT is the highest, up to 226 °C. The densities of DAPOP and DAPOT measured by densimeter at room temperature are 1.831 and 1.819 g·cm^−3^, respectively, which are higher than those of RDX. The friction sensitivities of DAPOP and DAPOT are estimated to be 224 N and 360 N, respectively, and the impact sensitivities are 24 J (DAPOP) and 40 J (DAPOT), respectively. DAPOP and DAPOT also exhibited relatively excellent detonation performance. The detonation velocities are 8249 m·s^−1^ and 7865 m·s^−1^, respectively, and the detonation pressures are 30.0 GPa and 25.9 GPa, respectively. The results show that the energetic salts of DAPO have good detonation performance and safety, which is helpful for the design and exploration of new energetic materials with high energy density for application in the field of high-energy and insensitive energetic materials.

## Data Availability

The data presented in this study are available in the paper. The CCDC 2160045, 222315, 222316, and 222317 contain supplementary crystallographic data for this paper. These data can be obtained free of charge via http://www.ccdc.cam.ac.uk/conts/retrieving.html (accessed on 1 December 2022) (or from the Cambridge Crystallographic Data Centre, 12, Union Road, Cambridge CB2 1EZ, UK; fax: +44 1223 336033).

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
