# Peer review of "Synthesis, Crystal Structure, and Characterization of Energetic Salts Based on 3,5-Diamino-4H-Pyrazol-4-One Oxime"

_molecules, 2023, doi:10.3390/molecules28010457_

Round 1

Reviewer 1 Report

The presented manuscript reported a detailed investigation of energetic salts based on 3,5-diamino-4H-pyrazol-4-one oxime. The salt structures, as well as the combination design of anions and cation are interesting to researchers in the field energetic materials. However, a major revision is necessary for the manuscript itself. Some detailed comments are as follows:

1. A more comprehensive introduction and references are needed to further clarify the advantages of DAPO type structures. For example:

https://doi.org/10.1016/j.fpc.2021.09.003

https://doi.org/10.1002/jhet.3434

2. The thermal behaviors of the energetic salts are very different. Authors should give more explanations on this issue rather than simply list the results. A comparative thermal study of the thermal properties of the compounds is also suggested.

3. In Figure 5, the author should make sure all the expressions are in English.

4. Based on the performance of the compounds, the authors are suggested to give some information of their potential applications.

5. The writing of this manuscript is a little rough. A more careful language polishing is highly suggested.

6. Considering the existed skeleton of DAPO is the basis of this work, more and deep discussion of the synthesis of DAPO, such as the exploration of synthetic mechanism or improvement of synthetic method, is suggested to be added in the manuscript.

Author Response

Thanks so much for your letter and comments concerning our manuscript entitled “Synthesis, Crystal structure, and Characterization of Energetic Salts Based on 3,5-Diamino-4H-Pyrazol-4-One Oxime” (NO.: molecules-2117560). We thank the reviewers for the time in closely viewing our manuscript. And also thank the reviewers for giving us insightful and constructive suggestions. These comments are very helpful for revising and improving our manuscript. If the opportunity is there, we are so much looking forward to receive your guidance to our work. After carefully studying the reviewers’ comments, we have made corresponding changes to the manuscript which we hope meet with your approval. Words presented RED color are the changes we have made in the revised manuscript. We hope that our manuscript would be published in this prestigious journal.

The main corrections in the revised manuscript and the responses to the editor and reviewer’s comments are as flowing:

Reviewer 1

  1. A more comprehensive introduction and references are needed to further clarify the advantages of DAPO type structures. For example:

https://doi.org/10.1016/j.fpc.2021.09.003

https://doi.org/10.1002/jhet.3434

Response:We first thank you for your kind guidance of our paper and this will encourage us to carry out more meaningful research in the later scientific work. We carefully read all literatures abovementioned, which is very helpful to our paper. In order to improve our manuscript, we have carefully studied your valuable comments and made corresponding modifications in the revised manuscript. We hope the new version can meet with your approval.

  1. The thermal behaviors of the energetic salts are very different. Authors should give more explanations on this issue rather than simply list the results. A comparative thermal study of the thermal properties of the compounds is also suggested.

Response:Your suggestion is greatly appreciated and adopted. The thermal stability of DAPOT·3H2O is higher than that of DAPOP·H2O is because DAPOT·3H2O has more hydrogen bonds and stronger π-π interactions. Since the anion of DAPOC·H2O is not energetic ion, its exothermic peak is gentler and releases less energy than other energetic salts based on DAPO.

  1. In Figure 5, the author should make sure all.

Response:Thanks so much for your comment. The Figure 5 was replaced by new version with the expressions are in English.

  1. Based on the performance of the compounds, the authors are suggested to give some information of their potential applications.

Response::Your kind advice and comments for our manuscript were very much appreciated. The potential application of these compounds was provided. Especially,DAPOT has a high decomposition temperature, low mechanical sensitivity, and acceptable energy performance, and is an insensitive energetic material with excellent comprehensive performance.

  1. The writing of this manuscript is a little rough. A more careful language polishing is highly suggested.

Response:Thank you for your kind comments. Inspired by your comments, to make the logic and ideas of the article clearer than previous and then improve the manuscript, we have revised the whole manuscript including English (such as grammatical errors, inconsistent expressions, and some not correct sentences), description for the analysis results, and the appropriate format. Especially the English, we employed a native English speaker who engaged in English teaching work and revised the English writing of this manuscript carefully. All changes were highlighted using RED color words according to the requirements of the journal. We hope that the new presentation can describe the main work clearly and be recognized by Reviewers and Editor.

  1. Considering the existed skeleton of DAPO is the basis of this work, more and deep discussion of the synthesis of DAPO, such as the exploration of synthetic mechanism or improvement of synthetic method, is suggested to be added in the manuscript.

Response:Thank you so much for your precious comments. The optimized synthetic method of DAPO was carried out and discussed in the revised manuscript systematically.

Reviewer 2 Report

In this work, the authors synthesized DAPO-based energetic salts (DAPOP, DAPOC, DAPOT) through the reaction of DAPO with three acids including hydrochloric acid, Perchloric acid and Picric acid. The structures of the three salts have been verified by infrared spectroscopy, mass spectrometry, elemental analysis, and single crystal X-ray diffraction. The noncovalent interactions, thermal stability, density, enthalpy of formation and the detonation performance of the salts were considered. The motivation of the work and the approach adopted are well. The manuscript is publishable after the below mentioned questions/comments are addressed.

Comments:

1) Scheme 1: Synthesis energetic salts of DAPO Synthesis of energetic salts of DAPO

2) How do you calculate the standard molar enthalpy of formation (ΔfHθm) for the DAPOP and DAPOT Compounds? Please bring the equation.

3) Why don’t you calculate the standard molar enthalpy of formation (ΔfHθm) for the DAPOC salt?

4) Please bring a reference for Gaussian 09 software used in this study.  

5) How sensitive are these materials to impact?

6) It would be useful to compare the measured properties of the synthesized compounds of DAPOP, DAPOC, DAPOT with nitrated-pyrazoles based energetic compounds reported in the literature.

Author Response

Thanks so much for your letter and comments concerning our manuscript entitled “Synthesis, Crystal structure, and Characterization of Energetic Salts Based on 3,5-Diamino-4H-Pyrazol-4-One Oxime” (NO.: molecules-2117560). We thank the reviewers for the time in closely viewing our manuscript. And also thank the reviewers for giving us insightful and constructive suggestions. These comments are very helpful for revising and improving our manuscript. If the opportunity is there, we are so much looking forward to receive your guidance to our work. After carefully studying the reviewers’ comments, we have made corresponding changes to the manuscript which we hope meet with your approval. Words presented RED color are the changes we have made in the revised manuscript. We hope that our manuscript would be published in this prestigious journal.

The main corrections in the revised manuscript and the responses to the editor and reviewer’s comments are as flowing:

Reviewer 2

1) Scheme 1: Synthesis energetic salts of DAPO→Synthesis of energetic salts of DAPO.

Response:Your kind advice and comments for our manuscript were very much appreciated. We have made changes in the corresponding Scheme 1.

2) How do you calculate the standard molar enthalpy of formation (ΔfHθm) for the DAPOP and DAPOT Compounds? Please bring the equation.

Response:Thank you so much for your precious comments. The the equation were provided in the supporting materials. Firstly, the enthalpy of formation of neutral molecule is required. The molecular formula of DAPO is C3H5N5O. According to the definition of enthalpy of formation, the enthalpy of formation of DAPO is equal to its enthalpy minus the enthalpy of the most stable element. The calculation formula is as follows:

(please see attachment)

The above enthalpy values are calculated quantitatively. All structures were optimized under B3PW91/6-31G (d, p), and then calculated under PWPB95D3/def2-QZVPP with high precision single point energy. After zero point energy correction, their enthalpy values were obtained respectively. Because the steady state of carbon in the standard state is the solid phase, the enthalpy of the gas phase is quantified, so the standard molar enthalpy of sublimation of the carbon needs to be added. Then, according to the enthalpy of formation of the neutral molecule, the enthalpy of formation of its cation M+ can be calculated, and the calculation formula is as follows:

(please see attachment)

Then, according to the empirical formula, the lattice energy of the ionic salt  was calculated:

/KJ mol-1=γ(ρm/Mm)1/3

ΔHL=+[p(ηM/2-2)+q(ηX/2-2)]/RT  (please see attachment)

Finally, according to the Born-Haber cycle, the enthalpy of formation of ionic salts can be calculated as follows:

(please see attachment)

The anions involved in the calculation of DAPOP and DAPOT are perchlorate and picrate, and their formation enthalpies are calculated using the G4 thermodynamic combination method in a process similar to the calculation of cation formation enthalpies. The enthalpy of formation of protons is derived from the National Institute of Standards and Technology (NIST).

3) Why don’t you calculate the standard molar enthalpy of formation (ΔfHθm) for the DAPOC salt?

Response:Thanks for your kind comments. Hydrochlorides of polynitrogen-containing energetic compounds are generally not energetic or have weak energy performance. This manuscript focuses on the energy performance of oxygenates such as perchlorate and picrate, so the enthalpy of formation and energy performance of DAPOC are not calculated.

4) Please bring a reference for Gaussian 09 software used in this study.

Response: Thanks for your kind recommendation. The reference for Gaussian 09 software has been added to the corresponding place

5) How sensitive are these materials to impact?

Response:Thank you so much for your precious comments. The impact and friction sensitivity were tested by standard BAM drop hammer and BAM friction tester respectively. DAPOP and DAPOT have low mechanical sensitivity (IS: 24-40 J; FS: 224-360 N) due to their layered stacking structures.

6) It would be useful to compare the measured properties of the synthesized compounds of DAPOP, DAPOC, DAPOT with nitrated-pyrazoles based energetic compounds reported in the literature.

Response:Your suggestion is greatly appreciated and adopted. Relative literatures were racked and searched. The energetic salts based on DAPO were compared with the energetic salts of the nitrated-pyrazoles based compound 4-oxyl-3,5-dinitropyrazolate. DAPO-based energetic salts have a higher density and enthalpy of formation,where DNPO-based energetic salts have good detonation performance due to their nitro and hydroxyl functional groups. DAPOT has a high decomposition temperature, low mechanical sensitivity, and acceptable energy performance, and is an insensitive material with excellent comprehensive performance.

Round 2

Reviewer 1 Report

The revised manuscript is acceptable and recommended to be pubished on Molecules.